

**Estimating Parameters in a Sea Ice Model using an Ensemble Kalman Filter**
**Yong-Fei Zhang[1, 2*], Cecilia M. Bitz[1], Jeffrey L. Anderson[3], Nancy S. Collins[3], Timothy J.**
**Hoar[3], Kevin D. Raeder[3], and Edward Blanchard-Wrigglesworth[1]**
[1]Department of Atmospheric Sciences, University of Washington, Seattle, Washington, USA.
[2]Now at Program in Atmospheric and Oceanic Sciences, Princeton University, Princeton, New Jersey,
USA.
[3]IMAGe, CISL, National Center for Atmospheric Research, Boulder, Colorado, USA.
Corresponding Author:
*Yong-Fei Zhang
Department of Atmospheric Sciences
Princeton University
4000 15th Ave NE
Seattle, WA 98195
USA
Phone: 1-512-298-9567
Email: yfzhang.nju@gmail.com
**Key points:**
•    Parameter estimation using an ensemble filter is done in a sea-ice model.
•    Parameters are improved during the data assimilation period.
•    Large improvements in model states are seen in the forecast period.







**Abstract**
Uncertain or inaccurate parameters in sea ice models influence seasonal predictions and
climate change projections in terms of both mean and trend. We explore the feasibility and
benefits of applying an Ensemble Kalman filter (EnKF) to estimate parameters in the Los
Alamos sea ice model (CICE). Parameter estimation (PE) is applied to the highly influential dry
snow grain radius and combined with state estimation in a series of perfect model observing
system simulation experiments (OSSEs). Allowing the parameter to vary in space improves
performance along the sea ice edge compared to requiring the parameter to be uniform
everywhere. We compare experiments with both PE and state estimation to experiments with
only the latter and found that the benefits of PE mostly occur after the DA period, when no
observations are available to assimilate (i.e., the forecast period), which suggests PE's relevance
for improving seasonal predictions of Arctic sea ice.













## 1. Introduction


Arctic sea ice has undergone rapid decline in recent decades in all seasons (e.g., *Stroeve et al.*,
2012;*Serreze and Stroeve*, 2015). The frequent large deviations of Arctic sea ice cover from its
climatology and the impact of sea ice cover on the overlying atmosphere and on ocean-
atmosphere fluxes motivates including an active sea ice component in seasonal to sub-seasonal
(S2S) weather forecasts (*Vitart et al.,* 2015). The persistence and reemergence of sea ice
thickness (SIT) and SST anomalies are major sources of predictability for Arctic sea ice extent
(*Blanchard-Wrigglesworth et al.*, 2011). Previous studies have demonstrated the importance of
accurate initial conditions, especially SIT, in predicting Arctic sea ice extent (*Day et al.*, 2014).
Hence studies applying data assimilation (DA) techniques to fuse observations with model
simulations are growing (e.g., *Lisæter et al.*, 2003; *Chen et al.*, 2017; *Massonnet et al.*, 2015),
most of which are focused on improving model states only, not the parameters in the sea ice
component.
Sea ice models, like other components of earth system models, can suffer large uncertainties
originating from uncertain parameters. The widely used Los Alamos sea ice model version 5
(CICE5), given its various complex schemes, has numerous uncertain parameters, such as in the
delta-Eddington shortwave radiation scheme (*Briegleb and Light,* 2007). The default values of
these parameters are usually chosen based on point-scale measurements that are taken on multi-
year sea ice (*Light et al,* 2008). *Urrego-Blanco et al.* (2015) conducted an uncertainty
quantification study of CICE5 and ranked the parameters based on the sensitivities of model
predictions to a list of parameters. This work provides guidance on which parameters could be
estimated using an objective method and during which seasons. Their findings suggest that the
estimates of the Arctic sea ice area and extent are especially sensitive to certain parameters (e.g.,



snow conductivity and snow grain size) in summer. However, they also discussed that their
sensitivities could be low as a consequence of prescribing atmospheric forcing in their model
setup, so parametric uncertainties are expected to be larger year round (particularly in winter) in
a fully-coupled model. Since we also run stand-alone CICE5 given that our aim is to demonstrate
the utility of parameter estimation (PE) for sea ice, we target the summer season.
Despite the importance of sea ice model parameters, few studies have tried to estimate or
reduce the parametric uncertainties, partly due to the large effort and computational cost if
parameter calibration is done in a trial-and-error fashion. A more systematic way is through DA.
*Anderson* (2001) demonstrated the feasibility of updating parameters using an ensemble filter in
a low-order model. *Annan et al.* (2005) was among the first to apply an ensemble filter to
estimate parameters in a complex earth system model. *Massonnet et al.* (2014) employed the
ensemble Kalman filter (EnKF) in a sea ice model to estimate three parameters that control sea
ice dynamics. In addition to achieving their goal of improving the sea ice drift, they also realized
slight improvements in the SIT distribution and extent as well as in the sea ice export through the
Fram Strait.
Our purpose is to expand upon previous studies to explore the feasibility of optimizing sea
ice parameters by asking how different observations (concentration and thickness in this study)
would constrain the parameters differently, whether we need to allow parameters to vary
spatially, and what are the benefits of the updated parameters both when observations are
available for assimilation (the DA period) and when observations are not available (the forecast
period).



Our sea ice DA framework is introduced in Section 2. Experimental design and metrics used
to evaluate model results are described in Section 3. We present results and discussions in
Section 4 and conclude in Section 5.

**2.  The sea ice data assimilation framework**
We use CICE5 linked to the data assimilation research testbed (DART) (*Anderson et al.*,
2009) within the framework of the Community Earth System Model version 2 (CESM2)
(http://www.cesm.ucar.edu/models/cesm2). The ocean is modeled as a slab ocean and the
atmospheric forcing is prescribed from a DART/CAM ensemble reanalysis (*Raeder et al.*, 2010).
Details of this framework can be found in *Zhang et al.* (2018). We extend DART/CICE to
include parameter estimation in this study. During the assimilation, DART and CICE5 cycle
between a DA step with DART and a one-day forecast step with CICE5. The state vector sent
from CICE5 to DART is augmented by adding selected sea ice parameters, so that when this
augmented state vector is passed into the filter during the DA step, the parameters and state
variables are both updated in the same way. The updated state variables are then post-processed
(if needed) and sent with the updated parameters back to CICE5 for the next one-day forecast
step. Unlike state variables, the parameters are not modified during CICE5 forecast steps.

**3.  Experiment design and evaluation methods**
We selected a tunable parameter in the Delta-Eddington solar radiation parameterization
treatment (*Briegleb and Light*, 2007), $R_{snw}$, to be estimated in this study. $R_{snw}$ represents the
standard deviation of dry snow grain radius that controls the optical properties of snow and is
one of the key parameters that determine snow albedo. Instead of directly tuning snow albedo



that could result in inconsistencies with the rest of the parameterization scheme, tuning $R_{snw}$
changes the inherent optical properties of snow in a self-consistent fashion (*Briegleb and Light,*
2007). Increasing $R_{snw}$ leads to smaller dry snow grain radius and larger snow albedo (*Hunke et*
*al.,* 2015). The default value of $R_{snw}$ is 1.5, which corresponds to a fresh snow grain radius of
125$\mu m$ (*Holland et al.,* 2012). Many parameters in CICE5, like $R_{snw}$, have default values based
on limited field observations. As sea ice models increase in complexity, empirical parameters
will increasingly need to be calibrated objectively.

The configurations of conducted experiments are listed in Table 1. We begin with a free run

of CICE5 without DA (hereafter FREE) with 30 ensemble members. Each ensemble member has
a unique value of $R_{snw}$, which is constant in time and space. The ensemble of $R_{snw}$ values were
random draws from a uniform distribution spanning -2 and 2. One of the ensemble members was
designated as the truth with the true value of $R_{snw}$. Following *Zhang et al.* (2018), synthetic
observations were created by adding random noise to sea ice concentration and thickness (SIC
and SIT, respectively) taken from the truth ensemble member. The noise follows a normal
distribution with zero mean and a standard deviation of 15% for SIC and 40 cm for SIT. The
FREE experiment does not assimilate any observations, and the $R_{snw}$ values stay the same
throughout the experimental period.

We then conducted two pairs of experiments to test the feasibility of estimating parameters

using the Ensemble adjustment Kalman filter (EAKF) (*Anderson*, 2002), which is a deterministic
ensemble square root filter. Each experiment assimilates daily SIC or SIT synthetic observations.
The first pair is referred to as DAsicPEcst and DAsitPEcst, while the second is referred to as
DAsicPEvar and DAsitPEvar. In each pair, the former assimilates SIC observations and the latter
SIT observations. In the first pair, each ensemble member has a unique spatially-uniform $R_{snw}$. In



the second pair, we allow a separate value of $R_{snw}$ at each horizontal grid point. The augmented
state has the single parameter for $R_{snw}$ in the first pair or the two-dimensional grid of $R_{snw}$
parameters in the second pair.

All variables in the sea ice state vector are two-dimensional in space. The parameter $R_{snw}$ and

the state variables were updated based on their correlations with neighboring observations. The
posterior ensemble generated by DART is always spatially varying. For the first pair of
experiments, we take an area-weighted average of the two-dimensional posterior to get a
spatially invariant $R_{snw}$ to send back to CICE5. For the second pair of experiments, the spatially
varying posterior $R_{snw}$ was sent to CICE5. In all experiments, the sea ice component was run for
a day to produce a new state that was augmented with the previous times posterior $R_{snw}$ (which is
not prognostic in CICE5) for the next DA cycle. To increase the prior ensemble spread of $R_{snw}$, a
spatially and temporally adaptive inflation was applied to the priors of both the model states and
$R_{snw}$ before they were sent to the filter (*Anderson*, 2007). The initial value, standard deviation,
and inflation damping value of the adaptive inflation are 1.0, 0.6, and 0.9.  The localization half-
width is 0.01 radians (about 64 km) as discussed in *Zhang et al.* (2018). We also reject
observations that are three standard deviations of the expected difference away from the
ensemble mean of the forecast.

A third pair of experiments was conducted with only state DA (no parameter estimation),

known as DAsic and DAsit, that assimilate daily SIC and SIT synthetic observations,
respectively. DAsic and DAsit have the same ensemble set of $R_{snw}$, which is also the initial set of
$R_{snw}$ in the above PE experiments. The ensemble of $R_{snw}$ remains fixed throughout the
experiment period.



All experiments begin on 1 April 2005 and run for 18 months. Synthetic observations are
assimilated only during the first 6 months (the DA period), and the next 12 months are a pure
forecast period to mimic the real-world situation when making a forecast. The values of $R_{snw}$ are
unchanged once DA ceases. We chose not to utilize DA beyond October 2005 for two reasons.
First, sea ice states have small ensemble spread in winter, as illustrated in Figure 1a, so DA
updates tend to be small. In contrast, the relatively larger spread from April to October ensures
that assimilating observations can have more impact in updating model state variables and
parameters. Second, the snow albedo feedback only influences the sea ice state when sunlight is
present.
Several commonly used error indices were calculated to evaluate the performance of the
experiments. The temporal averaged root-mean-square error ($RMSE_s$) and the area weighted
spatial averaged root-mean-square error ($RMSE_t$) are defined as follows:
$$RMSE_s = \sqrt{\frac{\sum_{i=1}^{N}(\overline{x_i^m}-x_i^t)^2}{N}}; \; RMSE_t = \sqrt{\frac{\sum_{j=1}^{M}(\overline{x_j^m}-x_j^t)^2}{M}}$$

where $i$ and $j$ are the indices in time and space, $x$ may refer to parameters or model states, $N$ is
the number of days and $M$ is the number of grid cells. The superscripts $m$ and $t$ refer to model
and truth, respectively. The overbar indicates the mean of the model ensemble.
Model bias is defined as the mean of the 30 member ensemble of the experiments minus the
truth. Absolute bias difference (ABD) between two experiments is defined as follows:

$$ABD = \left|\overline{x_i^{case1}}-x_i^t\right| - \left|\overline{x_i^{case2}}-x_i^t\right|$$

where $x$ may refer to parameters or model states, the superscripts $t$ refers to the truth, *and case1*
and *case2* refer to the two experiments to compare. The overbar indicates the mean of the model
ensemble. RAB indicates how much improvement or degradation DA offers relative to the
control (FREE) run.



## 4. Results and Discussion

4.1 Temporally and spatially invariant parameters

The ensemble mean of FREE underestimates SIC throughout the year (Figure 1a) partly because our arbitrary ensemble member selected as the truth has an above average $R_{snw}$ (Figure 1c). As such, we would intuitively expect $R_{snw}$ to have a positive increment as a result of assimilating SIC observations. Figure 1b confirms that $R_{snw}$ increments are positive, with the posterior ensemble mean gradually approaching the true value during the DA period in the spatially-constant PE experiments (DAsicPEcst and DAsitPEcst). The posterior $R_{snw}$ has smaller ensemble spread than the prior $R_{snw}$ (also see Figure S1d, e, and f), which is consistent with the EAKF theory. In Figure 1c DAsitPEcst outperforms DAsicPEcst starting in June, indicating that SIT provides more information than SIC for $R_{snw}$. Similarly, with state-only DA, *Zhang et al.* (2018) found that SIT is more efficient than SIC observations at constraining state variables. There could be several reasons why the rate at which $R_{snw}$ approaches the true value decreases with time. First, the ensemble spread of $R_{snw}$ may be insufficient because no uncertainty is introduced into $R_{snw}$ in CICE5 during the forecast step. It is an open question how much additional uncertainty should be introduced into the parameters. To help avoid filter divergence, we apply the prior adaptive inflation to the parameters (as well as to the model states), which may still be not enough.   Second, the correlation between $R_{snw}$ and the observations may be too weak. Solar radiation becomes very low by the end of September and hence $R_{snw}$ has little impact on sea ice, which explains the weak correlation between $R_{snw}$ and the observations (further discussed below).  Either reason could result in a negligible update to $R_{snw}$.

The correlations between $R_{snw}$ and the observations have unique spatial patterns and evolve with time. On May $1^{st}$, the correlation between $R_{snw}$ and SIC is generally positive (Figure 2a).



The positive correlations are significant especially where SIC is under ~100%. Larger $R_{snw}$
corresponds to higher snow albedo and more reflected sunlight, which in turn delays the melting
of sea ice. The correlations are still significant along the ice edges in August (Figure 2c) and
become noisier and have less significant values by the end of the melt season (Figure 2e). The
correlation between $R_{snw}$ and SIT has different spatial patterns (Figures S2b, S2d, and S2f).
Negative correlations between $R_{snw}$ and SIT on May 1$^{st}$ can be seen in the Chukchi Sea, Beaufort
Sea, and East Siberian Sea, where $R_{snw}$ and SIC have positive correlations. This suggests that
where SIC increases with $R_{snw}$ in spring, it is possible that SIT actually decreases, which might
be due to elevated concentration raising the compressive strength and reducing sea ice
deformation. While a brighter surface is able to reduce thickness over large regions in spring, the
effect is mostly gone by the end of summer when positive correlation prevails.

4.2 Spatially varying $R_{snw}$
We discussed in section 4.1 that processes relating $R_{snw}$ and observed quantities have
complex spatial features. The spatial map of the posterior $R_{snw}$ and the reduction in the ensemble
spread of $R_{snw}$ after EAKF in the first pair of experiments (Figure S1) also suggest that the
updates are concentrated on the ice marginal zones. It may be too crude to use a single value of
$R_{snw}$ for the whole Arctic. We let $R_{snw}$ be a spatially varying parameter in the second pair of PE
experiments, even though the true $R_{snw}$ is spatially invariant. The spatial features of $R_{snw}$ will
purely depend on how $R_{snw}$ correlates with the observations. As in DAsicPEcst and DAsitPEcst,
the analysis field of $R_{snw}$ is spatially varying, and we did a spatial averaging to get a single
number for the next run. $R_{snw}$ along the sea ice edges get updated more, while $R_{snw}$ in the center
is less influenced. But the averaging smoothed out this spatial feature. In DAsicPEvar and





DAsitPEvar, we didn't do spatial averaging at the end of each DA cycle, but let the spatially
varying 2D field of $R_{snw}$ be the $R_{snw}$ field in the next run, so the spatial feature was carried along
the simulation.
Figure 3 depicts the ABD of $R_{snw}$ (defined in section 2) between different pairs of
experiments at the end of the DA period.  Figures 2a and 2d confirm that DAsicPEcst and
DAsitPEcst improve the $R_{snw}$ comparing to FREE. Figures 2b and 2e show the spatial feature of
improvements or degradations in $R_{snw}$ for the two spatially varying PE experiments. They both
show the contrast between the ice marginal zones and the central Arctic. Improvements are
mostly seen along the ice edges. Spotty improvements in the inner Arctic can be found in
DAsitPEvar (Figure 3e), while degradations are prevailing in the inner Arctic in DAsicPEvar
(Figure 3e). Figures 2c and 2f highlight the improvements or degradations from allowing $R_{snw}$ to
vary spatially. The general features are that DAsicPEvar and DAsitPEvar have reduced $R_{snw}$
biases more along the ice edges compared with DAsicPEcst and DAsitPEcst. However,
degradations (Figure 3c) or negligible improvements (Figure 3f) are found in the central Arctic.
This suggests that spatially invariant PE generally works better for the whole pan-Arctic regions,
while spatially varying PE can work well in the ice marginal zones but not in the central Arctic,
especially when SIC is the only observed quantity.  SIC has little variability in the central Arctic
and hence assimilating the SIC observations will not add much information for parameters or
model states. The degradations in $R_{snw}$ but slight improvements in SIC (discussed in section 4.3)
in the central Arctic suggest that $R_{snw}$ is likely over adjusted to cancel out other errors (e.g., noise
from atmospheric forcing fields).

4.3 Additional improvements in model states





We demonstrated that $R_{snw}$ approaches the true value by assimilating SIC or SIT (at different
rates) in the previous sections. We now investigate whether PE also improves the simulation of
model states, beginning with timeseries of the pan-Arctic sea ice area and volume in all of our
experiments (see Figure 4).
In our preceding work, we showed that assimilating SIC and SIT could improve model
states (*Zhang et al.*, 2018), which can also be confirmed in Figure 4. During the DA period,
DAsic can efficiently reduce biases in area, but DAsic has limited influence on volume. Within
about a month into the forecast period, DAsic improves neither area nor volume. In contrast,
DAsit is highly beneficial at reducing both area and volume during the DA period, with at least
some improvement to volume persisting through the whole 1-year forecast period.
We find that updating $R_{snw}$ has a relatively large impact on volume beginning in spring of
the forecast period (Figure 4b). Either treating $R_{snw}$ as a spatially varying or constant parameter
has about the same effect until late summer of the forecast period. In fact, all of the PE
experiments outperform the state-only DA experiments in the forecast period. As shown in Table
1, SIT DA with PE always performs the best, reducing the bias in area by up to 63% and
reducing the bias in volume by up to 73%. SIC DA with PE is second best in terms of simulating
the area, reducing the bias by up to 37%. SIC DA with PE is comparable to DAsit in simulating
volume, reducing the bias by around 30%.
Finally, we compare the spatial patterns of bias reduction in SIC and SIT from PE
experiments by comparing RMSE of SIT in DAsicPEcst and DAsitPEcst to their state-only DA
counterparts, DAsic and DAsit (see Figure 5). The comparisons are made in two periods: the DA
period (April to October 2005) and the forecast period (April to September 2006). *Zhang et al.*
(2018) showed that the DAsic could only improve SIT along the sea ice edges. Figure 5a

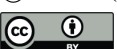



demonstrates that DAsicPEcst offers some improvements in the central Arctic as well.
Improvements resulted from a more accurate $R_{snw}$ in the forecast period are more prominent
(Figure 5b). For DAsitPEcst, SIT is improved almost everywhere in the Arctic, with slight
degradations along the ice edges (Figure 5c). The improvements persist throughout the forecast
period (Figure 5d).

5. **Conclusions**

We extend the functionality of DART/CICE to do parameter estimation (PE) through the

EAKF as well as updating the model states. One of the key parameters determining sea ice
surface albedo, $R_{snw}$, is estimated as an example in this study. $R_{snw}$ is updated using the filter. We
designed a series of perfect model observing system simulation experiments (OSSEs) to
demonstrate the feasibility of PE in CICE5. Results show that $R_{snw}$ gradually approaches the true
value during the data assimilation (DA) period (from April to October 2005). Updating
parameters with PE could further improve the model state estimation but not prominently in the
DA period. During the forecast period, with a better representation of the parameter, the PE
experiments show significant superiority over the state-only DA experiments, both in SIC and
SIT. The results in the forecast period indicate that by updating parameters as well as state
variables, assimilating SIC observations only is comparable to assimilating SIT observations. We
concluded that SIT is the most important variable to be observed in *Zhang et al.* (2018), but
satellite observations of SIT have large uncertainties and only cover a short time period. We
could alternatively improve model parameters by assimilating SIC observations with the ultimate
goal of improving SIT. Results from the subset of experiments treating $R_{snw}$ as a spatially
varying parameter suggest that the $R_{snw}$ biases are mostly reduced along the sea ice edges but not



as much in the central Arctic. We suggest that varying $R_{snw}$ spatially is not effective when
conducting DA for the whole Arctic, but worth exploring when it comes to regional studies, such
as in the seasonal sea ice zones.

**Acknowledgements**
This work was supported by the National Oceanographic and Atmospheric Administration
Climate Program Office through grant NA15OAR4310161. We thank Adrian Raftery and
Hannah Director for helpful discussions, and David Bailey and Marika Holland for suggestions
about choosing the proper parameters to estimate in the Los Alamos sea ice model. We
acknowledge Computational & Information Systems Lab at the National Center for Atmospheric
Sciences and Texas Advanced Computer Center at The University of Texas at Austin for
providing high performance computing resources that have contributed to the research results
reported within the paper. The model outputs archiving is underway and will be available in the
figshare repository.











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



Table 1. List of experiments with different configurations and RMSE of the total Arctic sea ice
area and volume calculated over two experiment periods: DA (April to October, 2005) and
forecast (April to September, 2006) for the seven experiments. All the experiments use the same
localization half-width and prior inflation algorithm as stated in section 3.

| Experiments | Observations assimilated | Parameter estimate | RMSE of Arctic sea ice area $(10^6 km^2)$ | | RMSE of Arctic sea ice volume $(10^3 km^3)$ | |
|---|---|---|---|---|---|---|
| | | | DA | Forecast | DA | Forecast |
| FREE | None | None | 0.250 | 0.343 | 0.711 | 1.302 |
| DAsic | SIC | None | 0.120 (-52%) | 0.345 (4%) | 0.583 (-18%) | 1.285 (-1%) |
| DAsicPEcst | SIC | Spatially constant | 0.114 (-55%) | 0.217 (-37%) | 0.520 (-27%) | 0.887 (-32%) |
| DAsicPEvar | SIC | Spatially varying | 0.123(-51%) | 0.240(-30%) | 0.601 (-16%) | 1.130 (-13%) |
| DAsit | SIT | None | 0.113(-55%) | 0.327(-5%) | 0.247 (-65%) | 0.868 (-33%) |
| DAsitPEcst | SIT | Spatially constant | 0.103 (-59%) | 0.141 (-59%) | 0.210 (-70%) | 0.349 (-73%) |
| DAsitPEvar | SIT | Spatially varying | 0.103 (-59%) | 0.129 (-63%) | 0.222 (-69%) | 0.376 (-71%) |








**Figure captions**
**Figure 1.** Time series of (a) the Arctic sea ice area and (b) sea ice volume from a CICE5 free run.
Each gray line represents one ensemble member, black line the ensemble mean, and red line the
truth. Time series of (c) the parameter Rsnw for two DA experiments. Blue line represents
DAsicPEcst that assimilates SIC observations, magenta represents DAsitPEcst that assimilates
SIT,  and green line the experiment DA_PAR_CST. The red reference line indicates the true
value of Rsnw. Each error bar represents two standard deviations of the 30 ensemble members of
Rsnw. Error bar is shown for every five days.

**Figure 2.** Correlations between (a) $R_{snw}$ and SIC and (b) $R_{snw}$ and SIT for 2005-05-01, (c) $R_{snw}$
and SIC and (d) $R_{snw}$ and SIT for 2005-08-01, and (e) $R_{snw}$ and SIC and (f) $R_{snw}$ and SIT for
2005-10-01. At each point, we calculate the correlation of $R_{snw}$ and the observed quantities
across the 30 ensemble members on the selected dates. The posterior states outputted from the
experiments DAsicPEcst and DAsitPEcst are used for calculation.

**Figure 3.** The differences of absolute mean bias (ABD, see Eq 2) of Rsnw between the DA
experiments: (a) DAsicPEcst, (b) DAsicPEvar, (d) DAsitPEcst, and (e) DAsitPEvar and the
control experiment FREE, and between the spatially-varying PE experiments and the spatially-
constant PE experiments: (c) DAsicPEvar and DAsicPEcst, and (f) DAsitPEvar and DAsitPEcst.

**Figure 4.** Daily biases of (a) the total Arctic sea ice area and (b) the total Arctic sea ice volume
for FREE (black), DAsic (blue), DAsicPEcst (green), DAsicPEvar (purple), DAsit (orange),
DAsitPEcst (pink), and DAsitPEvar(red). Gray dash line in each plot represents the zero



reference line. The blue line in (a) is overlapped by the purple and green lines in the first half of
time. The black line in (a) is overlapped by the orange and blue lines in the second half of time.
The black line in (b) is overlapped by the blue line from February to July.
**Figure 5.** The relative differences of RMSE of SIT between DAsicPEcst and DAsic for the (a)
DA experiment period and (b) forecast period, and between DAsitPEcst and DAsit for the (c)
DA experiment period and (d) forecast period. The differences of RMSE are divided by the
RMSE of DAsic and DAsit, respectively, to get the relative differences.

**Figure S1.** The posterior values of Rsnw for the experiment DAsitPEcst on (a) 2005-06-01, (b)
2005-08-01, and (c) 2005-10-01, and the differences between the ensemble spread of posterior
Rsnw and that of prior Rsnw (the posterior minus prior) for the experiment DAsitPEcst on (d)
2005-06-01, (e) 2005-08-01, and (f) 2005-10-01.










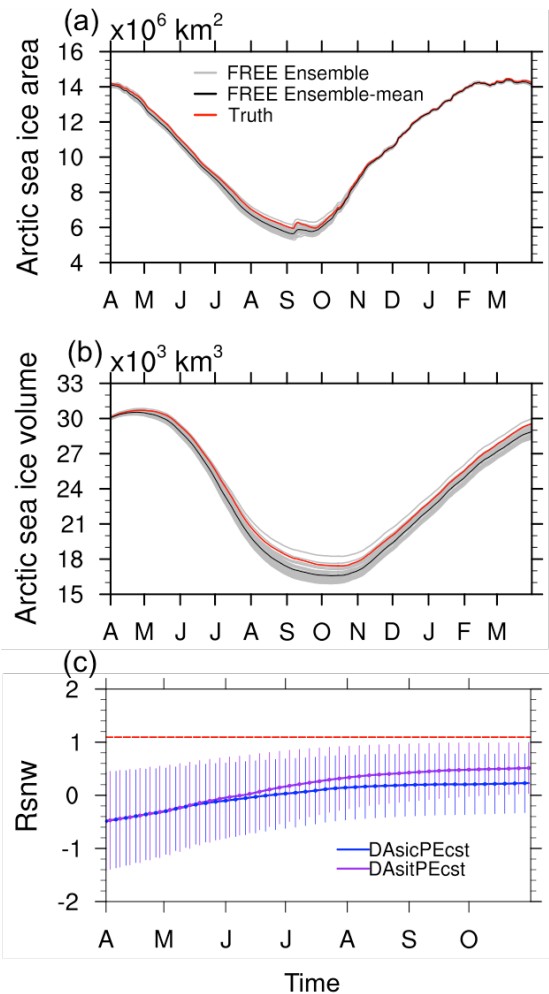


Figure 1. Time series of (a) the Arctic sea ice area and (b) sea ice volume from a CICE5 free

run. Each gray line represents one ensemble member, black line the ensemble mean, and red

line the truth. Time series of (c) the parameter $R_{snw}$ for two DA experiments. Blue line

represents DAsicPEcst that assimilates SIC observations, magenta represents DAsitPEcst that

assimilates SIT,  and green line the experiment DA_PAR_CST. The red reference line

indicates the true value of $R_{snw}$. Each error bar represents two standard deviations of the 30

ensemble members of $R_{snw}$. Error bar is shown for every five days.

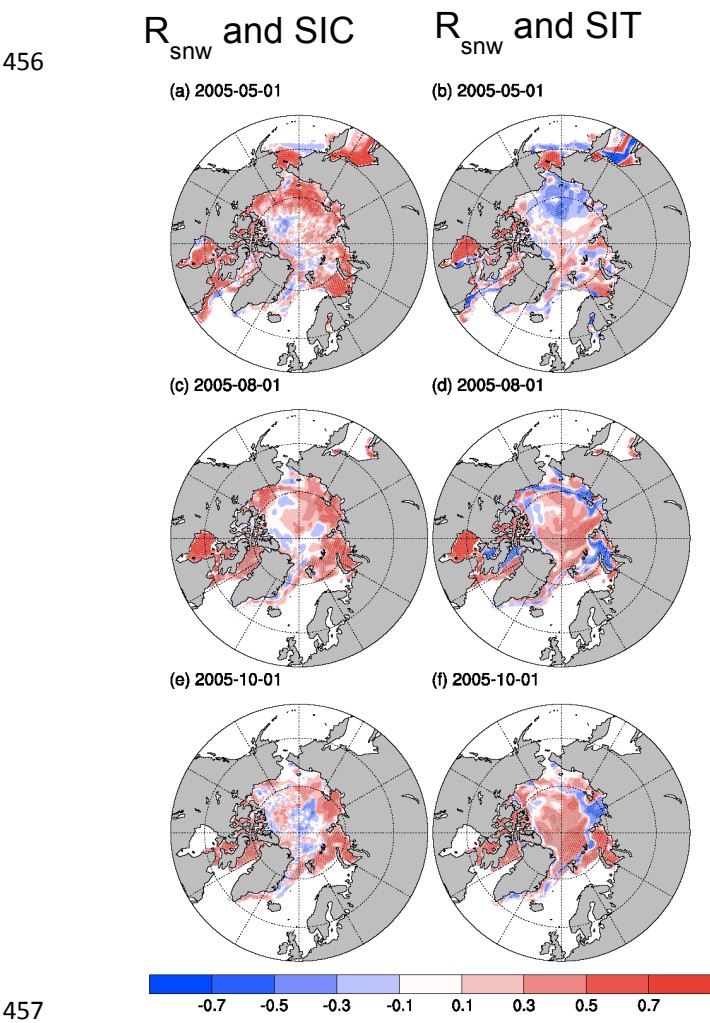

Figure 2. Correlations between (a) $R_{snw}$ and SIC and (b) $R_{snw}$ and SIT for 2005-05-01, (c) $R_{snw}$

and SIC and (d) $R_{snw}$ and SIT for 2005-08-01, and (e) $R_{snw}$ and SIC and (f) $R_{snw}$ and SIT for

2005-10-01. At each point, we calculate the correlation of $R_{snw}$ and the observed quantities

across the 30 ensemble members on the selected dates. The posterior states outputted from the

experiments DAsicPEcst and DAsitPEcst are used for calculation.


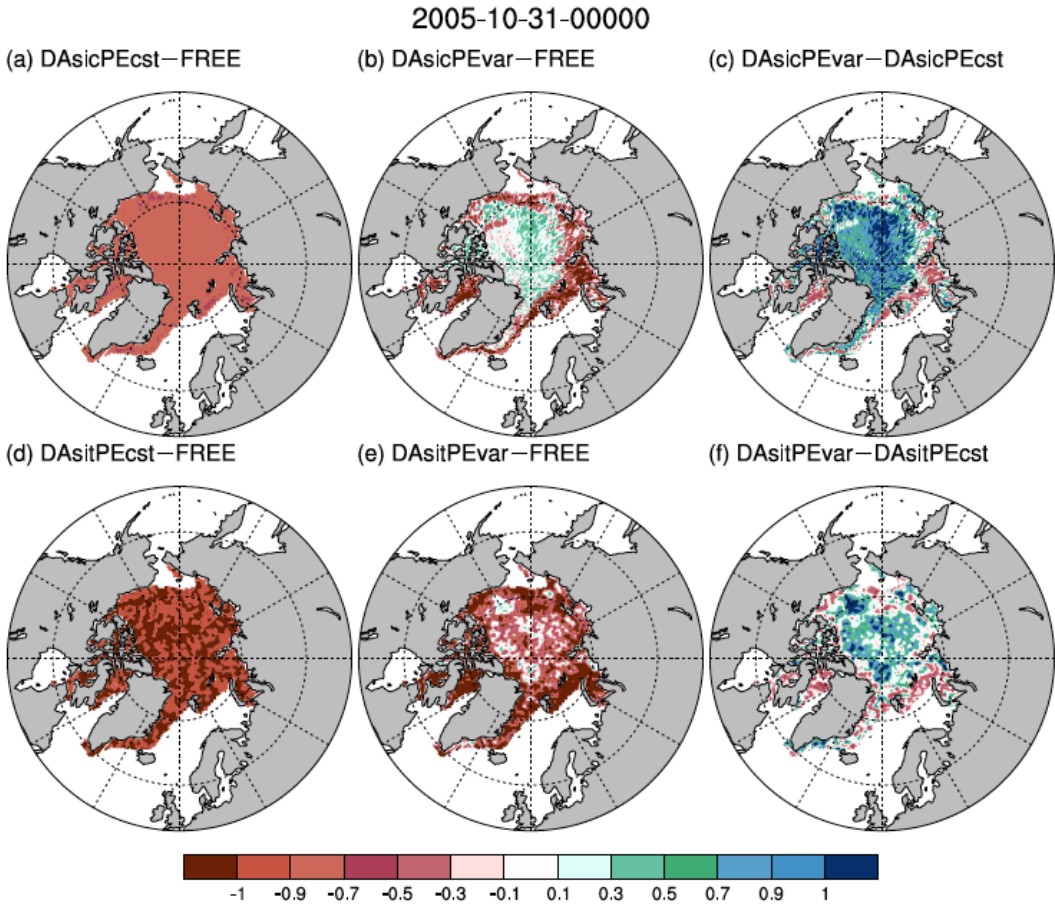

Figure 3. The differences of absolute mean bias (ABD, see Eq 2) of $R_{snw}$ between the DA
experiments: (a) DAsicPEcst, (b) DAsicPEvar, (d) DAsitPEcst, and (e) DAsitPEvar and the
control experiment FREE, and between the spatially-varying PE experiments and the spatially-
constant PE experiments: (c) DAsicPEvar and DAsicPEcst, and (f) DAsitPEvar and DAsitPEcst.



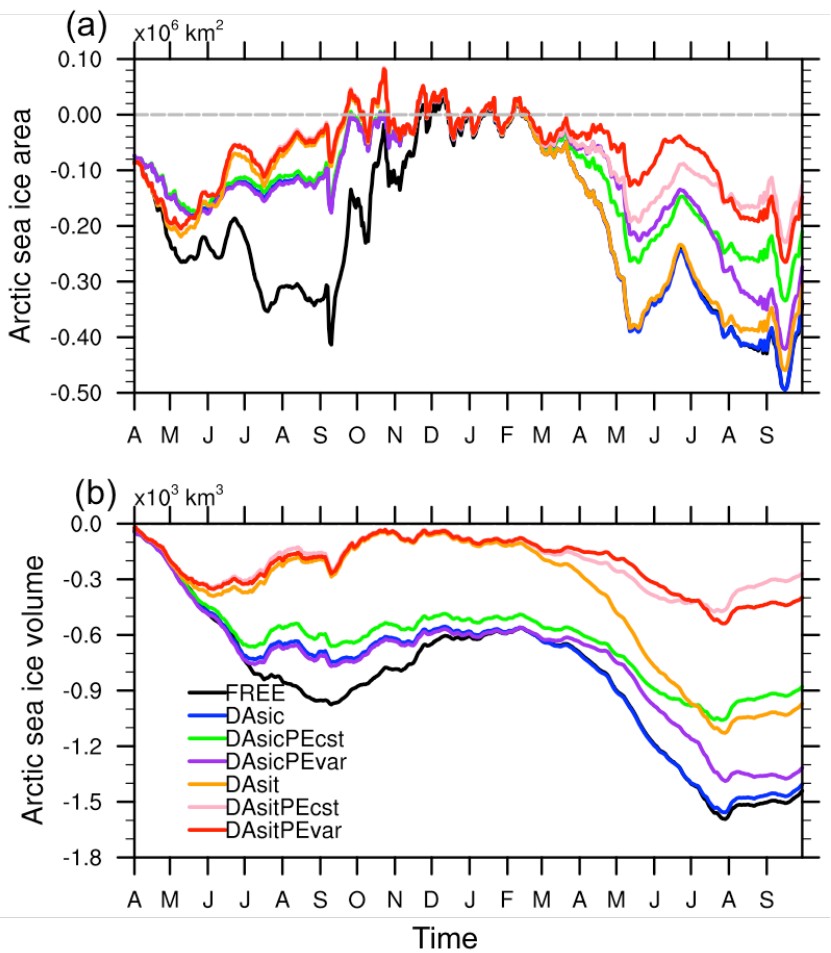


Figure 4. Daily biases of (a) the total Arctic sea ice area and (b) the total Arctic sea ice volume
for FREE (black), DAsic (blue), DAsicPEcst (green), DAsicPEvar (purple), DAsit (orange),
DAsitPEcst (pink), and DAsitPEvar(red). Gray dash line in each plot represents the zero
reference line. The blue line in (a) is overlapped by the purple and green lines in the first half of
time. The black line in (a) is overlapped by the orange and blue lines in the second half of time.
The black line in (b) is overlapped by the blue line from February to July.




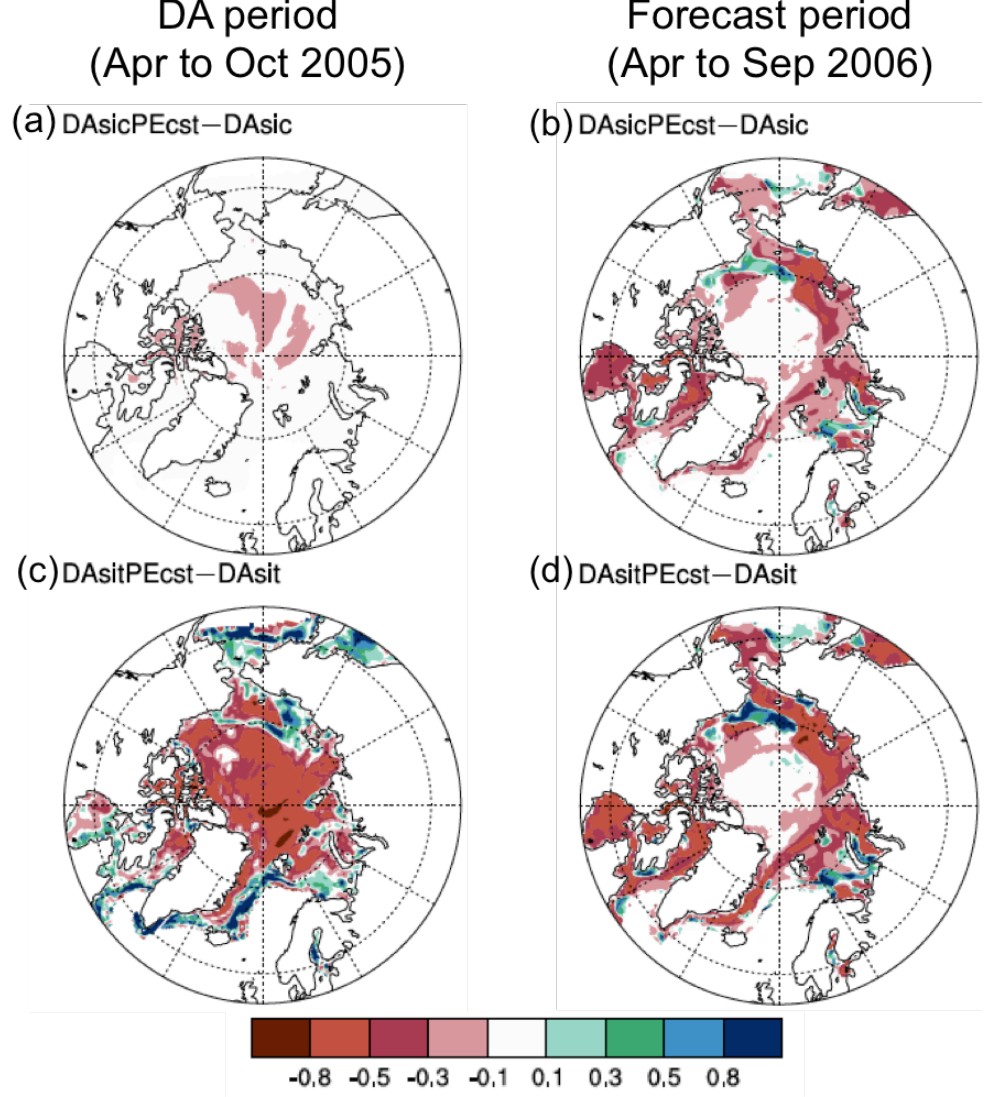

Figure 5. The relative differences of RMSE of SIT between DAsicPEcst and DAsic for the (a)
DA experiment period and (b) forecast period, and between DAsitPEcst and DAsit for the (c)
DA experiment period and (d) forecast period. The differences of RMSE are divided by the
RMSE of DAsic and DAsit, respectively, to get the relative differences.