# Peer review of "Estimating Parameters in a Sea Ice Model using an Ensemble Kalman Filter"

_The Cryosphere, 2020_

## Referee Comment (RC1) · Anonymous Referee #1 · 4 Jul 2020

This study utilizes a perfect model study with the sea ice model CICE5 and a Ensemble Kalman Filter in order to demonstrate the usefulness of varying a selected parameter. In this case the Snow grain size (Rsnw). The study investigate both a constant Rsnw and a Rsnw that varies in spaces. The spatially varying Rsnw improves the results near the sea ice edge but degrades the results in the central Arctic.

Results are based on a series of 18 month experiments that includes a data assimilation period of 6 month during summer as Rsnw only influences the results here.

A general note is that studies like these is valuable for calibration purposes, however with a model like CICE that is very complex it can be hard to extract one parameter and calibrate this without calibrating the entire model. This nicely outlined put in from line 215 to 220 where the author describes a potential less obvious mechanism that

causes a slightly unexpected result.

The study is conducted as a perfect model study which means that all state variables are available and the truth is known. Can this be transferred to a real observation? I would like to see some comments about this as for instance ice thickness based on altimetry is not available in summer, which is the period chosen for the calibration.

With some minor corrections I find the study worthwhile for publication

I would like the authors to check the figure references as they seems to point to wrong figures from time to time. Especially in the description of figure 3.

Abstract : I would like a comment on the variation of Rsnw vs the constant.

Line 39. Despite DA being a normal acronym for data assimilation I would write it in full potentially adding the short version. One should be able to read the abstract without reading the rest in order to find acronyms.

Line 64: Calibration of the none model state parameters are still calibrated in order to improve model state (in this case ice concentration ice thickness). I would rephrase this a bit.,

Line 80: The aim is to improve sea ice forecast all year (I would assume) but the parameter that is chosen is active in summer therefore it makes sense to focus on summer. A slight reformulation is desirable

Line 108. I assume that this is only Rsnw that is updated beside the state vector. This is mentioned later but I would like it to be here.

Line 127 – 164: I think that it would make it easier to read if you start describing the free run, then the data assimilation runs (constant Rsnw), and at last the experiments with varying Rsnw (either spatially constant or spatially varying).

Line 185 RAB?

Line 192. How does figure 1b show the positive increment of Rsnw? Is it 1c?

Line 238 Is it Figure 3a and 3d?

Linr 253 Any explanation for the ice thickness? This is lacking a bit.

Line 347. Is this a report? Can it be found?

Table 1: Two different RMSE's are defined in section 2. Which one is referred to here. Figure 1 The classical ice concentration/volume annual time series. The problematic part is that the variation from summer to winter is much larger than the variation between ensembles, truth and mean which is the interesting part. I think that it would make sense to normalize with the truth. I don't see the green line in the legend of c.

---

## Referee Comment (RC2) · Anonymous Referee #2 · 22 Jul 2020

Based on the OSSE framework, this paper extends the functionality of DART/CICE to do parameter estimation through the EAKF as well as updating the model states, and explored these impact on the simulation as well as the prediction of Artic sea ice. This study is systematic and well organized. However, I have some questions:

1. To avoid inconsistencies with the rest of the parameterization scheme, Rsnw is selected to be adapted via DA in this study. However, the snow conductivity is also important as mentioned in the introduction (Line 76). Why not tune snow conductivity through DA? In addition, Urrego-Blanco et al. (2015) suggests the interaction between Rsnw and snow conductivity, and how to consider this interaction in DA?

2. Although this study is based on OSSE, the simulated observations should mimic the real observations unless the goal of OSSE is to help evaluate the new observing

system. To our knowledge, the large scale SIT observations are mainly retrieved from satellites, while retrieval algorithms fail in the presence of water on the ice (e.g., SMOS and CryoSat-2). Thus, it is worth discussing whether assimilating SIT observation in summer is reasonable.

3. For SIV, the bias of DAsit is less than that of DAsicPEcst until 1 July 2006 (Fig. 4b). Hence, the conclusions drawn need to be more cautious, such as "The results in the forecast period indicate that by updating parameters as well as state variables, assimilating SIC observations only is comparable to assimilating SIT observations" (Lines 295-296).

4. Rsnw increments cannot be found in Fig. 1b (Line 192). Is it in Fig. 1c?

5. Green line cannot be found in Fig. 1 (Line 453).
* * *

---

## Author Response (AR2)

Dear Editor,

Thank you very much for coordinating the review process and giving general comments on the paper entitled "Estimating Parameters in a Sea Ice Model using an Ensemble Kalman Filter" by Yong-Fei Zhang et al. submitted to The Cryosphere. We've compiled the point-by-point response to referees' comments, general comments, and your final correction in the following text. All comments are listed, followed by our responses in bold.

Sincerely,
Yong-Fei Zhang, on behalf of co-authors

General comments:
* * *
* 329: Pls provide exact reference/link to a public accessible repository
for your data/information.

**Thanks for the comment. We've provided the link to a public repository for our data on the "data availability" section.**

* Naming:
Suggest to change "CICE5 free run" to "free CICE5 run" (442 & 486).
**Thank you for the suggestion. We've made changes accordingly.**

General comments:
* * *
* 76-80: Pls expand on the choice of summer as target season. During this time the processes in driving sea-ice processes are more complex than during the early growth season. Pls provide additional reasoning and evidence for this choice. (Give outlook to 168-173.) Include your comments on the suitability of ice-thickness products for summer, as these are typically non-trivial to derive, but especially not for summer.
**Thanks for the suggestion. We provided additional reasoning in line 86 as follows.**

**"Previous studies suggest that the ensemble spread of sea ice states is generally small in winter (e.g., *Lisaeter et al.*, 2003; *Fritzner et al.*, 2018; *Zhang et al.*, 2018), which will lead to limited update on model state variables or parameters. Also, sea ice concentration (SIC) reaches 100% in most of regions in winter and hence does not leave enough room for improvements by DA. The ensemble spread in summer, however, is much larger."**

**We also added a paragraph starting in line 93 to comment on SIT DA. The text is also copied below.**

**"Two types of observations are assimilated in our study, sea ice concentration and thickness (SIC and SIT, respectively). Satellite-retrieved SIC observations are widely utilized in the sea ice DA community, while the application of SIT observations is more challenging given its large uncertainty and lack of data in summer (*Zygmuntowska et al.*, 2014). Previous studies on Arctic sea ice predictability emphasized the importance of summer SIT observations (e.g., *Day et al.*, 2014; *Dirkson et al.*, 2017). We explore the benefits of SIT observations (in addition to SIC) on sea ice parameter estimation and advocate the needs of extending the data coverage of SIT observations into late spring and summer, which is actually possible in ICESat-2 (*Kwok et al.*, 2020). "**

* 101: Section 2.
Suggest to expand this with focus on DART. For example, what is implied with "extend" (line 106)?
Provide details on "(if needed)".
Provide more detail across all of section 2.
**Thanks for the suggestion. We've included more details in section 2. The following text was added in line 108.**

**"The default DART/CICE framework is only used for state estimation, we extend DART/CICE to include parameter estimation in this study."**

**The following text wad added in line 115.**

**"The post-process step is necessary when the updated variable goes beyond its physical boundaries, for example, when SIC is negative or larger than 100%."**

* 116: The movitation for choosing R_snw is not clearly demonstrated.

**The following sentence is added in line 124:**

**"We picked Rsnw because it is one of the parameters that the model predictions are sensitive to (*Urrego-Blanco et al.*, 2016) and is also one of the parameters perturbed to generate ensemble spread in *Zhang et al.* (2018)."**

* 125-126: Would you want to include further discussion on this, including an outlook on guidance to acquire observational data?
**Thanks for the suggestion. We believe more comprehensive observations of snow and ice properties, for example, the vertical profile of snow, would benefit more reliable representations of parameters in the model. The following text is added in line 160.**

**"More comprehensive observations at large scale will presumably benefit a better representation of snow and ice properties in sea ice models."**

* 132: Of major concern here, is the availability of sea-ice thickness information.
See above. This needs to be explored in the framework of which reliable and low uncertainty data are available.
**Thanks for the comment. We've added comments on the SIT DA. Please see our response above.**

* 434: "Figure S1": Missing from submitted manuscript.
--> Include in submission of revised ms.
**Thanks for the reminder. The Figure S1 is included in the revised manuscript.**

Specific comments:
* * *
39: Need to define "DA" at first use.
**Thanks, we've spelled it out.**

59: Define "SST" upon first use. - As only used once, suggest to replace "SST" with "sea-surface temperature".
**Thanks for the comment. It has been modified accordingly.**

63: Replace "growing" with "being investigated/developed" and rewrite the remainder of this sentence to improve your argument.
**We've modified the sentence in the text as follows.**

**"Hence studies applying data assimilation (DA) techniques to fuse observations with model simulations are actively investigated (e.g., *Lisæter et al.*, 2003; *Chen et al.*, 2017; *Massonnet et al.*, 2015), most of which are focused on improving model states only, not the parameters in sea ice parameterization schemes."**

66: Capitalize "earth", all through manuscript.
**Thanks for comment. We've capitalized "earth" throughout the manuscript.**

68: Rewrite "numerous uncertain parameters".
**We've changed it to "hundreds of uncertain parameters"**

70: Replace "point-scale" with "point".
**Done. Thanks.**

97-99: Suggest to remove this section.
**We've removed this section.**

109: Explain "augmented" for the given context.
**We've modified the text in line 117 as follows.**

**"During the DA step, the selected sea ice variables are placed into a "DART state vector" that is to be passed to the filter. The DART state vector is augmented by adding selected sea ice parameters, so that the parameters and state variables are both updated by the filter in the same way."**

118: Need to define "R_snw" at first use.
**Thanks for the comment. We rewrote the sentence as follows.**

**"The parameter we selected, Rsnw, represents the standard deviation of dry snow grain radius that controls the optical properties of snow and is one of the key parameters that determine snow albedo in the Delta-Eddington solar radiation parameterization treatment (Briegleb and Light, 2007)."**

168: Change "unchanged" to "held constant".
**We've changed the text as suggested.**

168: Rewrite "We chose not to utilize DA".
**We changed the phrase to "We do not perform DA".**

185: Correct "RAB" to "ABD".
**We've removed the incorrect sentence. Thanks for the comment.**

233: Poor English: "we didn't do spatial averaging at the end of each DA cycle,".
Suggest to change.
**The sentence has been modified in line 263 as follows.**

**"In DAsicPEvar and DAsitPEvar, we let the spatially varying 2D analysis field of Rsnw be the Rsnw field in the next run, so the spatial feature was carried along the simulation."**

289: OSSE already defined above: Replace "observing system simulation experiments (OSSEs)" with OSSEs.
**Thanks for the**

347-348: Provide proper reference for the CICE documentation.

**Thanks for the comment. The reference has been corrected as follows.**

**Hunke, E. C., W. H. Lipscomb, A. K. Turner, N. Jeffery, S. Elliott (2015), CICE: The Los Alamos Sea ice model documentation and software user's manual version 5, Los Alamos National Laboratory, Los Alamos, NM, USA, 116pp.**
This study utilizes a perfect model study with the sea ice model CICE5 and a Ensemble Kalman Filter in order to demonstrate the usefulness of varying a selected parameter. In this case the Snow grain size (Rsnw). The study investigate both a constant Rsnw and a Rsnw that varies in spaces. The spatially varying Rsnw improves the results near the sea ice edge but degrades the results in the central Arctic.

Results are based on a series of 18 month experiments that includes a data assimila- tion period of 6 month during summer as Rsnw only influences the results here.

A general note is that studies like these is valuable for calibration purposes, however with a model like CICE that is very complex it can be hard to extract one parameter and calibrate this without calibrating the entire model. This nicely outlined put in from line 215 to 220 where the author describes a potential less obvious mechanism that causes a slightly unexpected result.

The study is conducted as a perfect model study which means that all state variables are available and the truth is known. Can this be transferred to a real observation? I would like to see some comments about this as for instance ice thickness based on altimetry is not available in summer, which is the period chosen for the calibration.

With some minor corrections I find the study worthwhile for publication.

I would like the authors to check the figure references as they seems to point to wrong figures from time to time. Especially in the description of figure 3.

Thanks very much pointing it out. The figure references have been corrected.

Abstract : I would like a comment on the variation of Rsnw vs the constant.

**We've added a comment on the results from the spatially varying Rsnw experiments in the Abstract.**

**Relaxing the requirement that the estimated parameter be the same everywhere has benefits along the sea ice edge but degradations in the central Arctic, suggesting that spatially varying parameters will likely improve PE performance at local scales and should be considered with caution.**

Line 39. Despite DA being a normal acronym for data assimilation I would write it in full potentially adding the short version. One should be able to read the abstract without reading the rest in order to find acronyms.

**Thanks for the comment. We have spelled DA out.**

Line 64: Calibration of the none model state parameters are still calibrated in order to improve model state (in this case ice concentration ice thickness). I would rephrase this a bit.,

**We particularly refer parameters to those tunable parameters in the parameterization schemes, not model state variables. To clarify the point, we've changed 'the parameters in the sea ice component' to 'the parameters in sea ice parameterization schemes'.**

Line 80: The aim is to improve sea ice forecast all year (I would assume) but the parameter that is chosen is active in summer therefore it makes sense to focus on summer. A slight reformulation is desirable.

**By targeting summer we mean the DA experiments are done in summer but the forecast is for the full year. To make it clearer, we rephrase the sentence to 'we conduct DA experiments with PE in summer'.**

Line 108. I assume that this is only Rsnw that is updated beside the state vector. This is mentioned later but I would like it to be here.

**Thanks for the comment but we think it's fair to use general terms here since we are introducing the DA framework. The details of our experiments including the parameter to be tuned are described in Section 3.**

Line 127 – 164: I think that it would make it easier to read if you start describing the free run, then the data assimilation runs (constant Rsnw), and at last the experiments with varying Rsnw (either spatially constant or spatially varying).

**Thanks very much for the comment. We agree that it's clearer to describe experiments this way. We've modified the paragraph accordingly.**

Line 185 RAB?

**The typo is corrected. Thanks.**

Line 192. How does figure 1b show the positive increment of Rsnw? Is it 1c? Line 238 Is it Figure 3a and 3d?

**Sorry for making the confusions. All figure references are corrected.**

Linr 253 Any explanation for the ice thickness? This is lacking a bit.

**Thanks for the comment. We've added the following discussion on the SIT DA results.**

**Besides the improvements along the sea ice edges, the SIT DA also has benefit in the inner ice pack (Figure 3e), which is consistent with the results of the first pair of experiments that**

**SIT in general provides more information than the SIC observations, especially in the regions where SIC has little variability. However, spatially varying R$_{snw}$ has small advantages over spatially invariant R$_{snw}$ in the ice marginal regions but degradations in the central Arctic too (Figure 3f).**

Line 347. Is this a report? Can it be found?

**This refers to the CICE5 documentation. The reference has been corrected.**

Table 1: Two different RMSE's are defined in section 2. Which one is referred to here. Figure 1 The classical ice concentration/volume annual time series. The problematic part is that the variation from summer to winter is much larger than the variation be- tween ensembles, truth and mean which is the interesting part. I think that it would make sense to normalize with the truth. I don't see the green line in the legend of c.

**Yes we defined two RMSEs, one calculated over time and the other over space. The one calculated over time does not generate a spatial map so we dropped the subscript of RMSE$_s$ to clear up the confusion.**

**Thanks for checking the figure caption. Yes there's no green line, we've corrected the caption.**

**As for Figure 1, we thank the reviewer for kind suggestion but we think that the original plots are intuitive to show how Arctic sea ice area and volume evolve with season and it's straightforward to compare the true member with the rest of the ensemble members.**
Based on the OSSE framework, this paper extends the functionality of DART/CICE to do parameter estimation through the EAKF as well as updating the model states, and explored these impact on the simulation as well as the prediction of Artic sea ice. This study is systematic and well organized. However, I have some questions:

1. To avoid inconsistencies with the rest of the parameterization scheme, Rsnw is selected to be adapted via DA in this study. However, the snow conductivity is also important as mentioned in the introduction (Line 76). Why not tune snow conductivity through DA? In addition, Urrego-Blanco et al. (2015) suggests the interaction between Rsnw and snow conductivity, and how to consider this interaction in DA?

**Our study aims to demonstrate the feasibility of converging the ensemble of a parameter to its true value via sea ice data assimilation. We agree that there are other parameters worth exploring in the sea ice model, including snow conductivity and drag coefficients that will likely increase the model ensemble spread in winter as discussed. As you mentioned, it is tricky to factor in the interaction between different parameters, we need to proceed with caution updating multiple parameters. For example, if we want to create an ensemble of the snow conductivity parameter, shall we pair it with $R_{snw}$ for each ensemble member? If so, what's the correlation between Rsnw and snow conductivity? We believe those are interesting research questions worth exploring in our future work.**

2. Although this study is based on OSSE, the simulated observations should mimic the real observations unless the goal of OSSE is to help evaluate the new observing system. To our knowledge, the large scale SIT observations are mainly retrieved from satellites, while retrieval algorithms fail in the presence of water on the ice (e.g., SMOS and CryoSat-2). Thus, it is worth discussing whether assimilating SIT observation in summer is reasonable.

**Thanks for the comment. Yes the current SIT observations retrieved from satellites lack in the summer season. Other sea ice DA studies and seasonal predictability studies have suggested the importance of having SIT observations in late spring and summer, here we demonstrate that the SIT observations also provide more information for parameter estimation. Although we updated only one parameter in this study, we speculate the SIT observations would have more updates in most parameters than the SIC observations given the SIC variability is only large in ice marginal regions. So we'd like to advocate the needs of extending the coverage of SIT observations into late spring and summer, which is actually possible in ICESat-2 (Kwok et al., 2020).**

3. For SIV, the bias of DAsit is less than that of DAsicPEcst until 1 July 2006 (Fig. 4b). Hence, the conclusions drawn need to be more cautious, such as "The results in the forecast period

indicate that by updating parameters as well as state variables, assim- ilating SIC observations only is comparable to assimilating SIT observations" (Lines 295-296).

**Thanks for the comment. Our conclusion is only for the forecast period (from April to September) since seasonal sea ice forecasts normally won't start from winter. We agree that we need to make it more specific. We've added the forecast period in the text to clear up the confusion.**

4. Rsnw increments cannot be found in Fig. 1b (Line 192). Is it in Fig. 1c? 5. Green line cannot be found in Fig. 1 (Line 453).

**The figure references and captions are corrected. Thanks!**

**References**
**Kwok, R., Cunningham, G. F., Kacimi, S., Webster, M. A., Kurtz, N. T., & Petty, A. A (2020), Decay of the snow cover over Arctic sea ice from ICESat-2 acquisitions during summer melt in 2019. Geophysical Research Letters, 47, e2020GL088209. https://doi.org/10.1029/2020GL088209.**